# Detect Everything with Few Examples

**Xinyu Zhang**    **Yuhan Liu**    **Yuting Wang**    **Abdeslam Boularias**
{xz653, yl1834, yw632, ab1544}@rutgers.edu
Rutgers University

**Abstract:** Few-shot object detection aims at learning novel categories given only a few example images. It is a basic skill for a robot that performs tasks in open environments. Recent methods focus on finetuning strategies, with complicated procedures that prohibit a wider application. In this paper, we introduce DE-ViT, a few-shot object detector without the need for finetuning. DE-ViT's novel architecture is based on a new region-propagation mechanism for localization. The propagated region masks are transformed into bounding boxes through a learnable spatial integral layer. Instead of training prototype classifiers, we propose to use prototypes to project ViT features into a subspace that is robust to overfitting on base classes. We evaluate DE-ViT on few-shot, and one-shot object detection benchmarks with Pascal VOC, COCO, and LVIS. DE-ViT establishes new state-of-the-art results on all benchmarks. Notably, for COCO, DE-ViT surpasses the few-shot SoTA by 15 mAP on 10-shot and 7.2 mAP on 30-shot and one-shot SoTA by 2.8 AP50. For LVIS, DE-ViT outperforms few-shot SoTA by 17 box APr. Further, we evaluate DE-ViT with a real robot by building a pick-and-place system for sorting novel objects based on example images. The videos of our robot demonstrations, the source code and the models of DE-ViT can be found at https://mlzxy.github.io/devit.

**Keywords:** Robot Vision, Object Detection and Recognition, Few-shot Learning

## 1 Introduction

Object recognition and localization are two core skills of an autonomous robot operating in a new unstructured environment. *Few-shot object detection* is a promising approach for training a robot to detect novel categories based on a small set of support images [1]. However, most recent few-shot detection methods rely on *fine-tuning* on both base and novel classes [2], with complicated and tedious procedures that limit the practical use of these methods and that results in a large accuracy gap between the base and the novel classes [3]. Pretrained vision transformers (ViTs) [4, 5] can be used to overcome the limitations of fine-tuning. However, despite their rich semantical representations, pretrained ViT features lack the coordinates information that is required to perform a bounding box regression. As we show in Appendix C, naively applying a conventional regression on ViT features yields poor localization results, while unfreezing the ViT backbone leads to an accuracy collapse on novel classes, by completely overfitting the base classes.

To address these issues, we propose a novel localization architecture based on *region-propagation*. In this architecture, object proposals are expanded by a fixed ratio. Objects are localized by performing a mask prediction within the expanded proposals instead of a bounding-box regression. To accurately derive bounding boxes from the propagated regions, we propose the *spatial integral layer*, a learnable mask-to-box transformation. To further narrow the accuracy gap between base and novel classes, we propose to construct prototypes not as classifier weights, as shown in Fig. 2(a), but to project ViT features into a subspace that is used as network inputs. Our empirical studies demonstrate that the projected features are more robust to overfitting on base classes.

8th Conference on Robot Learning (CoRL 2024), Munich, Germany.

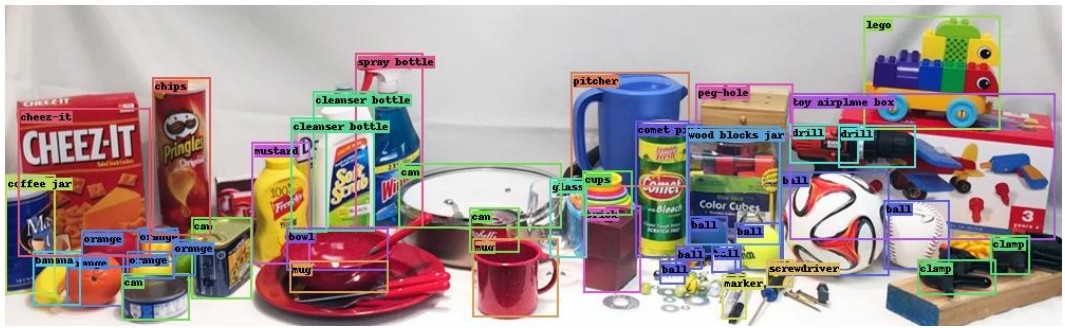

Figure 1: Demonstration of our method on YCB objects [6]. The model is trained on only the base categories of LVIS. A few example images of YCB objects are provided as novel categories during inference only.

With these proposed techniques, we introduce DE-ViT, a few-shot detector that uses example images to detect novel objects without the need for any finetuning or further training. An example of the results returned by DE-ViT is shown in Fig. 1. We evaluate DE-ViT on few-shot, and one-shot object detection benchmarks with Pascal VOC [7], COCO [8] and LVIS [9] datasets. DE-ViT establishes new state-of-the-art (SoTA) results on all benchmarks. For COCO, DE-ViT surpasses the SoTA LVC [10] by 15 mAP on 10-shot and by 7.2 mAP on 30-shot, and it also surpasses the one-shot SoTA BHRL [11] by 2.8 AP50. For Pascal VOC, DE-ViT surpasses the SoTA NIFF [12] by 2.0 nAP50. For LVIS, which has been regarded as a highly challenging dataset [13], DE-ViT outperforms the SoTA DiGeo [14] by 17 box APr. Notably, our method achieves a faster inference time while having a better accuracy. Further, we evaluate DE-ViT with a real robot in our novel-object sorting system.

## 2 Method

### 2.1 Problem Formulation

We use $\mathcal{C}$ to denote the set of classes. In few-shot object detection (FSOD), $\mathcal{C}$ is composed of a set of base classes, denoted by $\mathcal{C}_{base}$, and a set of novel classes, denoted by $\mathcal{C}_{novel}$. Thus, $\mathcal{C} = \mathcal{C}_{base} \cup \mathcal{C}_{novel}$ and $\mathcal{C}_{base} \cap \mathcal{C}_{novel} = \varnothing$. During training, a large number of examples are provided for the base classes. During testing, only $k$ labeled samples are provided for each novel class. The samples for novel classes are referred to as *support images*. The goal is to leverage the training data of $\mathcal{C}_{base}$ to learn a detector that can detect objects of $\mathcal{C}_{novel}$ given the $k$-shot support images.

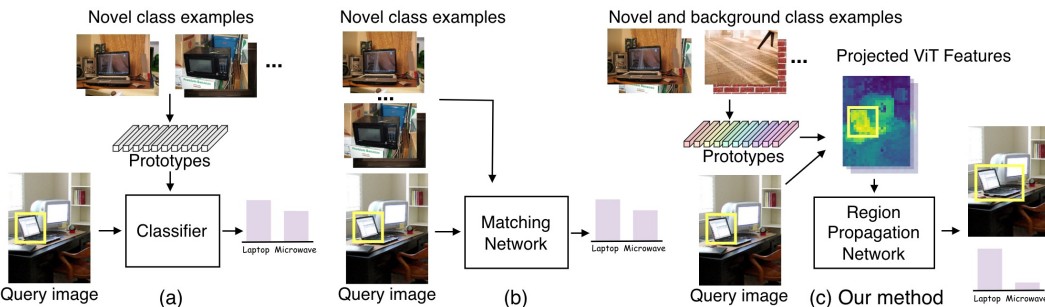

Figure 2: Existing meta-learning-based FSOD methods can be divided into two categories. Methods in the first category **(a)** build prototypes from novel class examples and use these prototypes as the classifier weights of a detection network. Despite its simplicity, this strategy exhibits inferior accuracy [2]. Methods in the second category **(b)** learn to match the proposal regions in the query image and novel examples through a matching network. This strategy is computationally heavy due to dense feature interactions across multiple images and usually requires finetuning to increase accuracy in novel classes [15, 16]. In contrast, our method **(c)** applies a dot-product with the prototypes to project ViT features into a subspace that is robust to overfitting on base classes, and then applies a region propagation network to refine the localization and derive the class score. Our method does not employ any finetuning for base or novel classes. Details of related work are in Appendix B.

## 2.2 Region Propagation Network

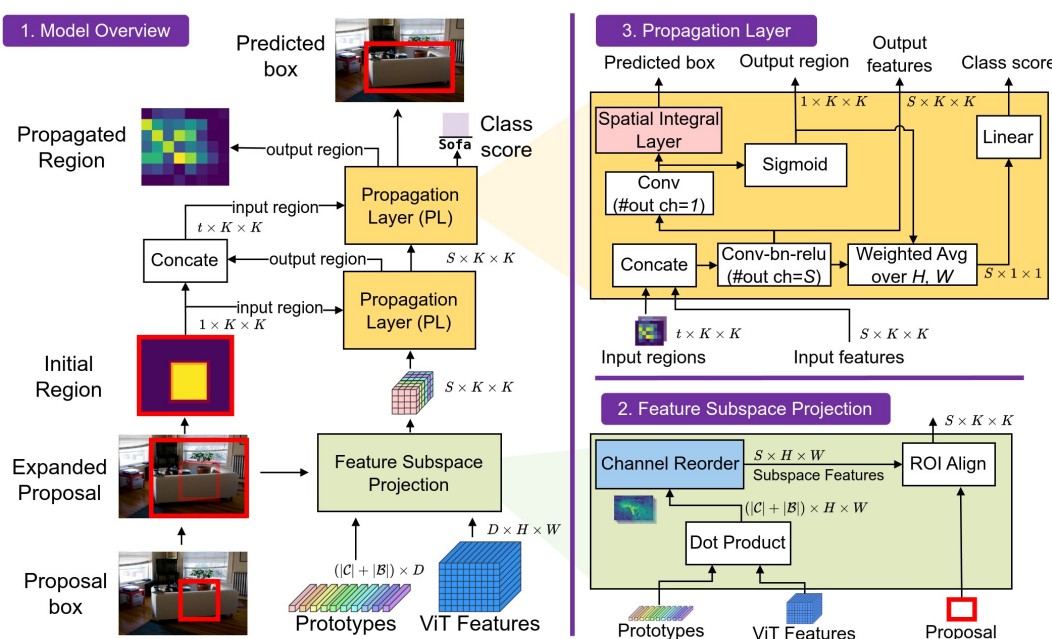

Figure 3: Overview of the proposed method. Given a proposal box of a query image, we extract the initial mask region (yellow) and expand the proposal by a constant ratio. Next, the region within the expanded proposal is gradually propagated and refined to fit the object area through a sequence of propagation layers. Each propagation layer accepts previous regions and features as input while returning updated features and region as outputs. The final predicted region is transformed to bounding-box coordinates through a learnable spatial integral layer, as detailed in Sec. 2.3. The predicted region also serves as spatial attention to average features along height and width. The averaged features are then mapped to class scores. The projected features are the dot products between the ViT features of a query image and class prototypes, as detailed in Sec. 2.4.

Despite having rich semantics information, pretrained ViT features lack the coordinates information required for bounding box regression. As shown in Appendix C, naively applying a conventional regression on ViT features yields poor localization results. A natural solution is to learn this localization capability by finetuning the ViT backbone during the training of the detector with the base classes. However, we observed that finetuning results in completely overfitting the base classes and in an accuracy collapse on novel classes. This was observed when integrating DINOv2 ViT into the framework of Meta RCNN [17], a standard prototype-based FSOD, as shown in Appendix C. This suggests that harnessing the generalization power of strong ViT backbones for FSOD is non-trivial. The question here is how to produce accurate localization with pretrained ViT features.

Given an object proposal, we use a region propagation network that gradually propagates the proposal region to accurately cover and fit the object by refining an object mask. Unlike bounding-box regression, mask prediction localizes objects without coordinate outputs. The propagated region is then transformed into a bounding box through a learnable spatial integral layer. We use an off-the-shelf region proposal network (RPN) to generate the initial region proposals, as class-agnostic proposals are shown to generalize well to novel classes [18]. Each proposal is expanded by a fixed ratio in order to delimit the propagation boundaries. The overall framework is shown in Fig. 3.

**Propagation Layer.** We propose the region *Propagation Layer* (PL), a new type of network module designed for object detection. PL serves as the central building block of our method. An example is shown in Fig. 5. The $t$-th PL block takes all previous regions $r_{0:t-1} \in \mathbb{R}^{t \times K \times K}$ and the previous PL block features $h_{t-1} \in \mathbb{R}^{S \times K \times K}$ as input, where $t$ denotes the number of PL blocks, $S$ denotes the number of feature channels, and $K$ denotes the feature spatial size. The $t$-th PL block outputs the updated region $r_t \in \mathbb{R}^{1 \times K \times K}$, features $h_t \in \mathbb{R}^{S \times K \times K}$, bounding box $b_t \in \mathbb{R}^4$, and class score

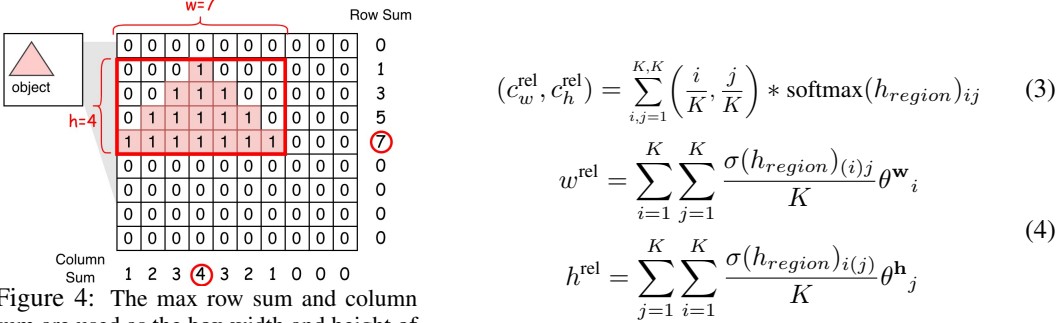

Figure 4: The max row sum and column sum are used as the box width and height of the triangular object.

$$(c_w^{\text{rel}}, c_h^{\text{rel}}) = \sum_{i,j=1}^{K,K} \left( \frac{i}{K}, \frac{j}{K} \right) * \text{softmax}(h_{region})_{ij} \quad (3)$$

$$w^{\text{rel}} = \sum_{i=1}^{K} \sum_{j=1}^{K} \frac{\sigma(h_{region})_{(i)j}}{K} \theta^{\mathbf{w}}_i$$

$$h^{\text{rel}} = \sum_{j=1}^{K} \sum_{i=1}^{K} \frac{\sigma(h_{region})_{i(j)}}{K} \theta^{\mathbf{h}}_j \quad (4)$$

$c_t \in \mathbb{R}$. Each PL block works as a small detection network and can be stacked to improve accuracy. The update rule is explained in Eq. 1 and illustrated in the third part of Fig. 3.

$$h_t = f_{\text{update},t}(\text{concat}(r_{0:t-1}, h_{t-1}); \theta), \ h_{t,region} = f_{\text{region},t}(h_t; \theta)$$
$$r_t = \sigma(h_{t,region}), \ b_t = f_{\text{integral},t}(h_{t,region}; \theta) \quad (1)$$
$$c_t = f_{\text{class},t}\big(\text{WeightedAvgPool}(h_t, r_t); \theta\big)$$

In Eq. 1, $f_{\text{update},t}$ denotes the conv-bn-relu block with $S$ output channels that updates the hidden features, and concat denotes channel-wise concatenation. $f_{\text{region},t}$ denotes the conv block with single channel output that predicts the output region logits $h_{t,region} \in \mathbb{R}^{1 \times K \times K}$. $\sigma$ denotes the sigmoid function. $f_{\text{integral},t}$ denotes the spatial integral layer detailed in Sec. 2.3. $h_t$ is aggregated over spatial dimensions to WeightedAvgPool$(h_t, r_t) \in \mathbb{R}^{S \times 1 \times 1}$ with weights $r_t$. $f_{\text{class},t}$ denotes the linear block that maps WeightedAvgPool$(h_t, r_t)$ to class scores. $\theta$ denotes network parameters. During training, we use focal loss and L1 regression loss for the output class score $c_t$ and bounding box $b_t$. For the output region $r_t$, we apply BCE loss and Dice loss [19]. The region labels during training are generated by the ground-truth object region within the expanded proposals.

## 2.3 Learnable Spatial Integral

Converting masks to bounding boxes is a widely-used transformation in instance segmentation networks [20] as a post-processing step. The standard solution is to find the top-left and bottom-right foreground pixels and use their positions as the bounding box coordinates [21]. However, this approach has major limitations. Firstly, this mask-to-box conversion assumes the availability of ground-truth instance masks, which are much more expensive to obtain than bounding boxes [8]. Moreover, this approach is non-differentiable and is also prone to outliers. Therefore, the question is how to accurately derive bounding boxes from the region-based localization results using a learnable and differentiable function.

Let $b^{\text{out}} = (c_w^{\text{out}}, c_h^{\text{out}}, w^{\text{out}}, h^{\text{out}})$ denote the output bounding box, where $c_w^{\text{out}} \in [0, W], w^{\text{out}} \in [0, W]$ and $c_h^{\text{out}} \in [0, H], h^{\text{out}} \in [0, H]$. Instead of predicting $b^{\text{out}}$ directly, we propose to first predict a relative bounding box $b^{\text{rel}} = (c_w^{\text{rel}}, c_h^{\text{rel}}, w^{\text{rel}}, h^{\text{rel}}) \in [0, 1]^4$, that can be transformed to $b^{\text{out}}$ according to Eq. 2,

$$(w^{\text{out}}, h^{\text{out}}) = (w^{\text{exp}} w^{\text{rel}}, h^{\text{exp}} h^{\text{rel}}),$$
$$(c_w^{\text{out}}, c_h^{\text{out}}) = (c_w^{\text{exp}} - 0.5w^{\text{exp}}, c_h^{\text{exp}} - 0.5h^{\text{exp}}) + (c_w^{\text{rel}} w^{\text{exp}}, c_h^{\text{rel}} h^{\text{exp}}), \quad (2)$$

where $b^{\text{exp}} = (c_w^{\text{exp}}, c_h^{\text{exp}}, w^{\text{exp}}, h^{\text{exp}})$ denotes the expanded proposal. Thus, $b^{\text{rel}}$ is a normalized bounding box relative to $b^{\text{exp}}$. Let $h_{region} \in \mathbb{R}^{K \times K}$ denote the output region logits, where we skip the notation of $t$-th block and the channel of 1 for simplicity. Our spatial integral layer $f_{\text{integral}}$ estimates $b^{\text{rel}}$ with Eq. 3 and 4. An illustrative example is given in Fig. 4.

To motivate Eq. 3 and 4, consider the toy example of converting a binary triangle mask to a bounding box in Fig. 4. A reasonable approach is to compute the mask center as the bounding box center and

Table 1: Results on COCO 2014 few-shot benchmark. Our method outperforms existing work in detecting novel classes by a significant margin. Results surpassing the SoTA are indicated in bold.

| Method | Requires Finetune | 10-shot | | | | 30-shot | | | |
|---|---|---|---|---|---|---|---|---|---|
| | | bAP | nAP | nAP50 | nAP75 | bAP | nAP | nAP50 | nAP75 |
| FSRW [22] | ✗ | - | 5.6 | 12.3 | 4.6 | - | 9.1 | 19 | 7.6 |
| Meta R-CNN [17] | ✗ | 5.2 | 6.1 | 19.1 | 6.6 | 7.1 | 9.9 | 25.3 | 10.8 |
| TFA [13] | ✓ | 33.9 | 10 | 19.2 | 9.2 | 34.5 | 13.5 | 24.9 | 13.2 |
| Multi-Relation Det [23] | ✗ | - | 16.6 | 31.3 | 16.1 | - | - | - | - |
| FSCE [24] | ✓ | - | 11.9 | - | 10.5 | - | 16.4 | - | 16.2 |
| Retentive RCNN [25] | ✓ | 39.2 | 10.5 | 19.5 | 9.3 | 39.3 | 13.8 | 22.9 | 13.8 |
| HeteroGraph [26] | ✓ | - | 11.6 | 23.9 | 9.8 | - | 16.5 | 31.9 | 15.5 |
| FsDetView [27] | ✓ | 6.4 | 7.6 | - | - | 9.3 | 12 | - | - |
| Meta Faster RCNN [15] | ✓ | - | 12.7 | 25.7 | 10.8 | - | 16.6 | 31.8 | 15.8 |
| LVC [10] | ✓ | 28.7 | 19 | 34.1 | 19 | 34.8 | 26.8 | 45.8 | 27.5 |
| CrossTransformer [16] | ✓ | - | 17.1 | 30.2 | 17 | - | 21.4 | 35.5 | 22.1 |
| NIFF [12] | ✓ | 39 | 18.8 | - | - | 39 | 20.9 | - | - |
| DiGeo [14] | ✓ | 39.2 | 10.3 | 18.7 | 9.9 | 39.4 | 14.2 | 26.2 | 14.8 |
| DE-ViT (Ours) ViT-S/14 | ✗ | 24 | **27.1** | **43.1** | **28.4** | 24.2 | **26.9** | 43.1 | **28.5** |
| DE-ViT (Ours) ViT-B/14 | ✗ | 28.3 | **33.2** | **51.4** | **35.5** | 28.5 | **33.4** | **51.4** | **35.7** |
| DE-ViT (Ours) ViT-L/14 | ✗ | 29.4 | **34.0** | **52.9** | **37.0** | 29.5 | **34.0** | **53.0** | **37.2** |

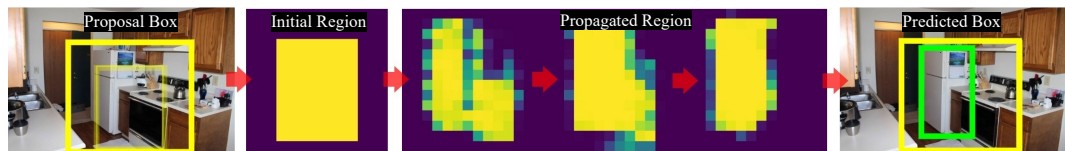

Figure 5: Region propagation on class "fridge". The proposal, expanded proposals, and final predicted boxes are colored in transparent yellow, yellow, and green, respectively. The three propagated regions (from left to right) are sampled from the output of the first three PL blocks in ascending order.

use the max row sum and column sum as width and height. Inspired by this insight, we compute the expected position of the spatial distribution softmax($h_{region}$) as the center of $b^{rel}$ in Eq. 3. We compute the row and column sums of the output region as $\sum_{j=1}^{K} \sigma(h_{region})_{ij}$ and $\sum_{i=1}^{K} \sigma(h_{region})_{ij}$ in Eq. 4. Instead of picking the maximum, we apply a soft aggregation to average all row or column sums in terms of magnitude rank. The aggregation is done by sorting the row and column sums, and then computing the weighted average. This explains the use of order statistics notation $(i)$ and $(j)$. Parameters $\theta^{h} \in \mathbb{R}^{K}, \theta^{w} \in \mathbb{R}^{K}$ are learnable aggregation weights.

## 2.4 Feature Subspace Projection

The main challenge of FSOD is to generalize to novel classes that are unseen during training. However, despite numerous attempts to solving this problem, by using margin-based regularization for example [14], there persists a considerable accuracy gap between base and novel classes [1]. This disparity indicates that a network trained with base classes would inevitably overfit on patterns that are only present among the base classes. A classic technique to reduce overfitting consists in representing data in a low-rank subspace [28]. We explore in this work the construction of a subspace of pre-trained ViT features that reduces the accuracy gap between base and novel classes.

Prototypes are class representatives built from support images. Given the support images of each class, we compute the ViT features, crop the features with object bounding boxes and use the average feature as the class prototype [17]. Let $p_{\mathcal{C}} \in \mathbb{R}^{|\mathcal{C}| \times D}$ denote the prototypes of classes from set $\mathcal{C}$, where $D$ denotes the channel dimension. Let $h_{vit} \in \mathbb{R}^{D \times H \times W}$ denote the ViT features of the query image. We assume that both prototypes and features are normalized to unit length at the channel dimension. Then $p_{\mathcal{C}} \cdot h_{vit} \in \mathbb{R}^{|\mathcal{C}| \times H \times W}$ can be interpreted as a subspace projection with $p_{\mathcal{C}}$ being the basis. However, this subspace construction has two limitations. Firstly, only using prototypes of classes set $\mathcal{C}$ can be too limited to sufficiently capture the feature information. Secondly, a permutation of $\mathcal{C}$ creates a different but equivalent subspace, yet designing permutation-invariant networks is a highly challenging problem [29]. For the first limitation, we introduce a set $\mathcal{B}$ of background classes, $\mathcal{B} \cap \mathcal{C} = \varnothing$, with $p_{\mathcal{B}} \in \mathbb{R}^{|\mathcal{B}| \times D}$ being the prototypes of $\mathcal{B}$, to preserve more information from $h_{vit}$. For the second limitation, we propose to build a separate subspace for each

Table 2: nAP50 results on Pascal VOC few-shot benchmark. Results surpassing the SoTA are indicated in bold. (*) denotes that implementation is not publicly available.

| Method | | Novel Split 1 | | | | | Novel Split 2 | | | | | Novel Split 3 | | | | | Avg |
|---|---|---|---|---|---|---|---|---|---|---|---|---|---|---|---|---|---|
| | | 1 | 2 | 3 | 5 | 10 | 1 | 2 | 3 | 5 | 10 | 1 | 2 | 3 | 5 | 10 | |
| TFA [13] | | 39.8 | 36.1 | 44.7 | 55.7 | 56.0 | 23.5 | 26.9 | 34.1 | 35.1 | 39.1 | 30.8 | 34.8 | 42.8 | 49.5 | 49.8 | 39.9 |
| FsDetView | | 25.4 | 20.4 | 37.4 | 36.1 | 42.3 | 22.9 | 21.7 | 22.6 | 25.6 | 29.2 | 32.4 | 19 | 29.8 | 33.2 | 39.8 | 29.2 |
| Multi-Relation Det [23] | | 37.8 | 43.6 | 51.6 | 56.5 | 58.6 | 22.5 | 30.6 | 40.7 | 43.1 | 47.6 | 31 | 37.9 | 43.7 | 51.3 | 49.8 | 43.1 |
| Retentive RCNN [25] | | 42.4 | 45.8 | 45.9 | 53.7 | 56.1 | 21.7 | 27.8 | 35.2 | 37.0 | 40.3 | 30.2 | 37.6 | 43 | 49.7 | 50.1 | 41.1 |
| Meta Faster R-CNN [15] | | 43.0 | 54.5 | 60.6 | 66.1 | 65.4 | 27.7 | 35.5 | 46.1 | 47.8 | 51.4 | 40.6 | 46.4 | 53.4 | 59.9 | 58.6 | 50.5 |
| LVC [10] | | 54.5 | 53.2 | 58.8 | 63.2 | 65.7 | 32.8 | 29.2 | 50.7 | 49.8 | 50.6 | 48.4 | 52.7 | 55 | 59.6 | 59.6 | 52.3 |
| CrossTransformer [16] | | 49.9 | 57.1 | 57.9 | 63.2 | 67.1 | 27.6 | 34.5 | 43.7 | 49.2 | 51.2 | 39.5 | 54.7 | 52.3 | 57 | 58.7 | 50.9 |
| HeteroGraph [26] | | 42.4 | 51.9 | 55.7 | 62.6 | 63.4 | 25.9 | 37.8 | 46.6 | 48.9 | 51.1 | 35.2 | 42.9 | 47.8 | 54.8 | 53.5 | 48.0 |
| DiGeo [14] | | 37.9 | 39.4 | 48.5 | 58.6 | 61.5 | 26.6 | 28.9 | 41.9 | 42.1 | 49.1 | 30.4 | 40.1 | 46.9 | 52.7 | 54.7 | 44.0 |
| NIFF [12] (*) | | **62.8** | **67.2** | 68.0 | 70.3 | 68.8 | 38.4 | 42.9 | 54.0 | 56.4 | 54 | 56.4 | 62.1 | 61.2 | 64.1 | 63.9 | 59.4 |
| DE-ViT (Ours) | ViT-S/14 | 47.5 | 64.5 | 57.0 | 68.5 | 67.3 | **43.1** | 34.1 | 49.7 | **56.7** | **60.8** | **52.5** | 62.1 | 60.7 | 61.4 | **64.5** | 56.7 |
| | ViT-B/14 | 56.9 | 61.8 | 68.0 | **73.9** | **72.8** | **45.3** | **47.3** | **58.2** | **59.8** | 60.6 | **58.6** | 62.3 | 62.7 | 64.6 | 67.8 | 61.4 |
| | ViT-L/14 | 55.4 | 56.1 | **68.1** | 70.9 | 71.9 | **43.0** | 39.3 | **58.1** | 61.6 | 63.1 | 58.2 | 64 | 61.3 | 64.2 | 67.3 | 60.2 |

class $c \in \mathcal{C}$, and reorder other classes $\mathcal{C} \setminus c$ to resolve permutation ambiguity. The feature subspace projection is explained in Eq. 5 and illustrated in Fig. 3.

$$h_{subspace,c} = \text{concat}(p_c \cdot h_{vit}, \text{channel-reorder}(p_{\mathcal{C}\setminus c} \cdot h_{vit}), p_{\mathcal{B}} \cdot h_{vit}) \tag{5}$$

In Eq. 5, $h_{subspace,c} \in \mathbb{R}^{S \times H \times W}$ denotes the subspace feature for class $c$, and function channel-reorder sorts the $|\mathcal{C}| - 1$ channels of input tensor $p_{\mathcal{C}\setminus c} \cdot h_{vit} \in \mathbb{R}^{(|\mathcal{C}|-1) \times H \times W}$ by magnitude at each spatial location, and then linearly interpolates the tensor to a pre-defined size $(S - 1 - |\mathcal{B}|) \times H \times W$, where $S$ is a constant hyperparameter. In practice, we use example images of non-object stuff classes, e.g., sky, road, floor, to construct $\mathcal{B}$. As shown in Appendix C, feature subspace projection significantly reduces the accuracy gap between base and novel classes. On the other hand, creating a subspace for each class $c \in \mathcal{C}$ introduces costly per-class inference. However, the per-class inference cost can be reduced by finding the top $T$ most likely classes with a lightweight prototype classifier [17] and only performing inference for these $T$ classes. As shown in Appendix C, our method achieves a faster inference speed and surpasses SoTA when $T = 3$.

## 3 Experiments

We comprehensively evaluate our method on few-shot and one-shot benchmarks. Furthermore, we compare the efficiency of our method against SoTA solutions, study few-shot performance for different numbers of shots, compare it to language-based detectors, and provide qualitative results. We conduct ablations to study the contributions of the proposed components to the performance of our method. Our source code and the pretrained models are included in the supplementary material and will be publicly released upon acceptance.

**Evaluation Metrics and Datasets.** Few-shot and one-shot evaluations split classes into base and novel classes. Base classes are seen during training and novel classes are unseen. The performance on novel classes is more important. For COCO and Pascal VOC, nAP, nAP50, and nAP75 represent mAP, AP50, and AP75 in novel classes. bAP and bAP50 represent mAP and AP50 in base classes. One-shot evaluation conventionally divides 80 classes of COCO into four even partitions, and alternatively uses three as base classes and one partition as novel classes [30]. There are 4 base/novel splits in total, named Split-1/2/3/4. For LVIS, APr, APc, and APf represent AP on rare, common, and frequent categories. Rare categories are used as novel classes. Metrics on LVIS, such as box APr and mask APr, are computed on bounding boxes and on instance segmentation masks separately. We evaluate our method on Pascal VOC [7], COCO [8], and LVIS-v1 [9]. We follow the conventional base/novel classes split and use the same support images of novel classes [13, 11].

**Implementation Details.** We adopt a standard two-stage object detection framework, similarly to Mask R-CNN [31]. We use the same off-the-shelf RPN with existing work [32] to generate object proposals. The RPN is trained separately for each dataset using only base classes. We use DINOv2 [33] ViT as the backbone, and report results in ViT-S/B/L (small, base, large) model sizes. The ViT backbones are kept frozen during detector training. We adopt the prototype extraction

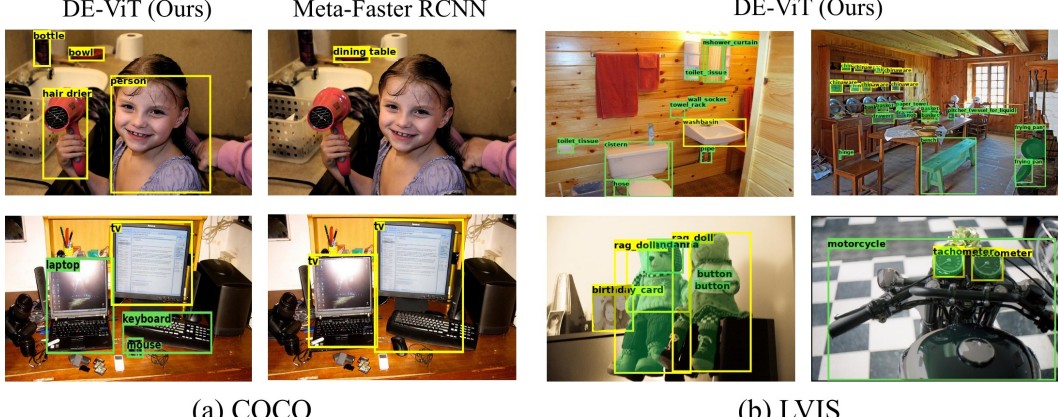

| DE-ViT (Ours) | Meta-Faster RCNN | DE-ViT (Ours) | |
|:---:|:---:|:---:|:---:|
| (a) COCO | | (b) LVIS | |

Figure 6: Qualitative visualization of our method DE-ViT on COCO (a) and LVIS (b), and comparison to Meta-Faster RCNN [15]. Boxes of base and novel classes are colored in green and yellow.

procedure of Meta RCNN [17] using ViT features. During training, the model can access only the prototypes of base classes. After training, the prototypes of novel classes are appended while the remaining parameters are unchanged. During evaluation, the model is evaluated on images that contain objects of both base and novel classes. Prototypes of background classes set $\mathcal{B}$ are extracted from images of stuff (non-object) classes, *e.g.*, sky, road, from COCOStuff [34] unless specified. Similar to [17], the top $T$ most likely classes for each proposal are determined by the distances between prototypes and the average proposal feature. We set $T$ to 10 unless specified, where $T$ is the number of classes to create subspace features and perform inference as explained in Sec. 2.4. We apply 3 PL blocks for experiments on Pascal VOC and COCO, and 5 PL blocks for those on LVIS.

### 3.1 Main Results

Tab. 1 shows our results on the few-shot COCO benchmark. Our method DE-ViT outperforms the previous SoTA LVC [10] by a significant margin (+15 nAP on 10-shot, +7.2 nAP on 30-shot). It is worth noting that LVC requires over ten stages for self-training and pseudo-labeling procedures on novel classes [35], while our method DE-ViT is trained once on the base classes and used directly on the novel classes. A pretrained model for LVC has never been released. We plot the nAP50 of our method and the SoTAs with different numbers of shots in Fig. 7.

Table 3: Performance comparison with existing FSOD methods on the LVIS dataset. We report the box AP.

| Method | | APr | APc | APf | AP |
|:---|:---|:---:|:---:|:---:|:---:|
| DiGeo [14] | | 16.6 | 22.8 | 28 | 24.4 |
| DE-ViT (Ours) | ViT-S/14 | **23.4** | 22.8 | 22.5 | 22.8 |
| | ViT-B/14 | **26.8** | **26.5** | 25.3 | **26** |
| | ViT-L/14 | **33.6** | **30.1** | 30.7 | 30.9 |

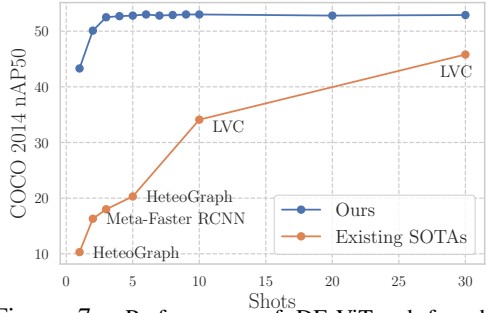

Figure 7: Performance of DE-ViT and few-shot SoTA with different numbers of shots.

Tab. 2 shows our results on the few-shot Pascal VOC benchmark. Our method DE-ViT outperforms the previous SoTA NIFF [12] by +2.0 on averaged nAP50. It is worth noting that the implementation of NIFF has not been released. LVIS has been regarded as a highly challenging benchmark for FSOD [13] with 337 novel classes and only DiGeo [14] reports few-shot results on LVIS v1. Tab. 3 shows that our method outperforms DiGeo in all metrics and a significant boost in the accuracy of detecting novel objects (+20 box APr).

Tab. 4 shows our results on the one-shot COCO benchmark. DE-ViT surpasses the previous SoTA BHRL by 6 bAP50 and 2.8 nAP50. One-shot detection task follows a single-class detection setting. Therefore, we adapt our method DE-ViT by detecting each class separately during evaluation. OWL-

Table 4: Results on COCO 2017 one-shot benchmark. DE-ViT outperforms existing work and is not limited to single class detection and single support image as other one-shot methods.

| | bAP50 | | | | | nAP50 | | | | |
|---|---|---|---|---|---|---|---|---|---|---|
| | Split-1 | Split-2 | Split-3 | Split-4 | Avg | Split-1 | Split-2 | Split-3 | Split-4 | Avg |
| SiamMask [30] | 38.9 | 37.1 | 37.8 | 36.6 | 37.6 | 15.3 | 17.6 | 17.4 | 17 | 16.8 |
| CoAE [37] | 42.2 | 40.2 | 39.9 | 41.3 | 40.9 | 23.4 | 23.6 | 20.5 | 20.4 | 22 |
| AIT [38] | 50.1 | 47.2 | 45.8 | 46.9 | 47.5 | 26 | 26.4 | 22.3 | 22.6 | 24.3 |
| SaFT [3] | 49.2 | 47.2 | 47.9 | 49 | 48.3 | 27.8 | 27.6 | 21 | 23 | 24.9 |
| BHRL [11] | 56 | 52.1 | 52.6 | 53.4 | 53.6 | 26.1 | 29 | 22.7 | 24.5 | 25.6 |
| DE-ViT (Ours, ViT-L/14) | **59.4** | **57.0** | **61.3** | **60.7** | **59.6** | 27.4 | **33.2** | **27.1** | **26.1** | **28.4** |

ViT [36] also reports one-shot results on COCO. However, OWL-ViT's results are obtained with an ensemble of language-based detection and one-shot pipelines without providing an implementation or isolated measurements. Therefore, we exclude OWL-ViT from the one-shot comparison.

## 3.2 Real Robot Experiment

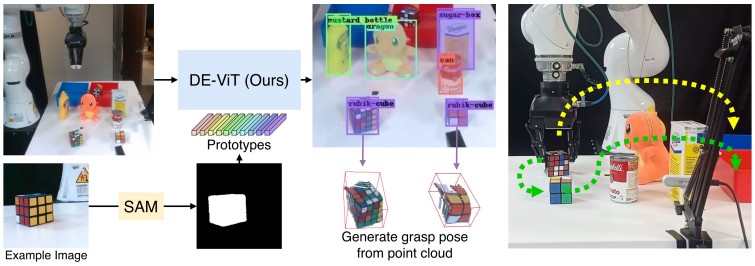

Figure 8: Overview of the system for sorting novel objects built upon our DE-ViT. Our system receives a few example images of novel classes and instantly detects new objects within the same category without tuning or further learning.

Table 5: Success rates of DE-ViT on sorting novel objects.

| Object | Success |
|---|---|
| Overall (%) | 97% |
| Chips | 8/10 |
| Tomato Can | 9/10 |
| Ball
Crackers
Brick
Cup
Mustard Bottle
Sugar Box
Orange
Cleanser Bottle | 10/10 |

**Setup.** To evaluate our DE-ViT in real-world robotic tasks, we develop a pick-and-place system for sorting novel objects based on example images. The system is outlined in Fig. 8 with an example of sorting Rubik's cubes. First, the front RGB camera takes images of the example objects, which are then segmented with SAM [39] and built as class representative prototypes. Next, our DE-ViT detects objects and generates instance segmentation masks from the RGB image of a side-view RGBD camera. Then, the instance segmentation masks are combined with the depth image to produce the point-cloud of each object instance. The grasp pose for each object is generated with a heuristic-based pose generator [40]. We use a Kuka LBR iiwa robot. Note that our system receives example images of novel classes and detects novel objects instantly as it does not require finetuning.

**Results.** Tab. 5 shows the success rates of our system in object sorting. We adopt ten YCB objects such as crackers and mustard bottles. For each run, all objects are placed on the table, and we ask the robot to pick and place objects based on a given order of the novel classes. We use our DE-ViT with the ViT-L backbone. We use 3 example images for each object class, except the mustard bottle and tomato soup can, for which we use 6 images to improve their detection accuracy. Our system achieves an overall 97% success rate out of 100 independent picks for all objects. The layout of all objects is randomized for each pick. Of the three failed cases, two involve mistaking chips for a tomato can and vice versa. In the third case, the chips are detected correctly, but the instance mask is not precise enough to enable a successful grasp. We believe this is because our DE-ViT model is trained exclusively on the base categories of the LVIS dataset without using YCB images or any datasets on retail products.

## 4 Final Remarks

We conducted extensive analyses and ablation studies on model efficiency, comparisons to DINOv2 in Meta RCNN, comparisons to language-based detectors [32], and the effects of the number of PL layers and shots. Details of these studies and results are included in Appendix C due to space limit.

## Acknowledgement

This work is partially supported by NSF awards 1846043 and 2132972.

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

# A  Appendix

## A.1  Code and Pretrained Models

The source code of our method DE-ViT is included in the supplementary folder `Code`. Please check `Code/README.md` for instructions on installation, dataset setup, and downloading pretrained models from an anonymous server.

# B  Related work

**Few-shot Object Detection (FSOD)** aims at detecting objects of novel classes by utilizing a few support images from novel classes as training samples [2]. Existing approaches can be broadly classified into finetuning-based [13, 25, 24, 27, 12] and meta-learning-based strategies [17, 22, 23]. Finetuning-based methods, despite their prevalence, suffer from a large accuracy gap between the base and novel classes, as well as practical limitations due to redundant multi-stage procedures [3]. Meta-learning-based methods avoid finetuning through online adaptation. Early works on meta-learning FSOD (Fig. 2.a) construct prototypes from novel class examples and use the prototypes as input to a network that classifies query images [17, 22]. Recent methods (Fig. 2.b) design interaction mechanisms of dense spatial features between query and support images [15, 16, 26]. In contrast, our method (Fig. 2.c) computes dot-products with the prototypes to project their features into a subspace, instead of using a prototype classifier, and applies a region-propagation network instead of a dense feature matching network. One-shot Object Detection (OSOD) is an extreme case of FSOD with only one example per novel class, it reduces the setting to single-class detection [11]. Prior approaches primarily focus on designing interaction mechanisms of dense spatial features between support and target images [30, 37, 41]. However, the OSOD formulation restricts the use of additional support and requires a separate inference per class [3, 38]. Compared with existing work, our method does not use per-class inference and only utilizes class-level prototypes without dense feature interactions.

**Vision Transformer (ViT)** is a recent architecture that demonstrates stronger performance and more interpretable features than traditional convolutional architectures. There is a growing trend of applying self-supervised ViTs, such as DINO [42, 33], to unsupervised object discovery [43, 44, 45, 5]. TokenCut [46] applies graph-cut over DINO features to separate the primary foreground object. DeepSpectral [4] predicts segmentation masks through unsupervised spectral clustering over DINO features. OW-DETR [47] uses DINO ViT to discover unknown objects in an open-world setting.

**Robotics Application of Few-shot Object Detection.** FSOD has been increasingly deployed in robotics applications [48]. AirInteraction [49] designs a robot exploration strategy based on human-informed interestingness, leveraging the *learning from examples* ability of few-shot detectors. CI-FSOD [50] designs a lightweight few-shot adaptor module that is suitable for robotics deployment on edge devices. Fewsol [51] studies the existing few-shot detector performance in typical robotics environments. TFOD [52] designs a mobile manipulator application purely based on the predictions of a few-shot detector with online annotations.

# C  Analysis and Ablation Studies

**Efficiency.** We compare the inference time of DE-ViT with different values of $T$ against recent few-shot works in Tab. A1 in COCO 10-shots setting, where Swin denotes Swin Transformer [53], RN101 denotes ResNet101 [54], MFRCNN denotes Meta Faster RCNN [15], and CrossT denotes Cross Transformer [16]. $T$ is the number of classes to create subspace features and perform inference. Tab. A1 shows that our method DE-ViT can achieve the smallest inference time while having better accuracy. The inference times of all the compared methods are measured on the same machine.

Table A1: Ablation study on different values of $T$ and inference time comparison with existing methods. N/A: The pretrained model is not publicly available for evaluation.

| Method | Backbone | $T$ | nAP50 | Secs/Img |
|--------|----------|-----|-------|----------|
| DE-ViT (Ours) | ViT-L/14 | 1 | **49.6** | **0.22** |
|  |  | 3 | **52.5** | **0.33** |
|  |  | 10 | **52.9** | 0.83 |
| LVC [10] | Swin-S | - | 34.1 | N/A |
| MFRCNN [15] | RN101 | - | 25.7 | 0.61 |
| CrossT [16] | Custom | - | 30.2 | 3 |

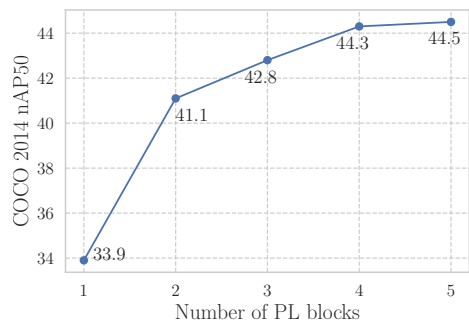

Figure A1: Ablation study of our method on different numbers of PL blocks with ViT-S/14 backbone.

Table A2: Comparison to naively applying DINOv2 with Meta RCNN, and ablation studies on the feature subspace projection.

| Meta RCNN + DINOv2 Backbone | Feature Subspace Projection | | | Novel | | Base | |
|---|---|---|---|---|---|---|---|
| | $c$ | $\mathcal{B}$ | $\mathcal{C} \setminus c$ | nAP50 | nAP75 | bAP50 | bAP75 |
| ✓ | | | | 4.5 | 2.2 | 48.9 | 22.5 |
| | ✓ | | | 26.2 | 9.7 | 29.3 | 12 |
| | ✓ | ✓ | | 38.4 | 23 | 43.4 | 26.8 |
| | ✓ | ✓ | ✓ | **39.5** | **24.1** | 42.3 | 25.9 |

**Feature subspace projection.** We examine the component effects for feature subspace projection in Tab. A2. The first column represents the integration of raw features of DINOv2 ViT into Meta RCNN [17], a standard prototype-based few-shot detector. Tab. A2 shows that the model has an accuracy collapse on novel classes with raw ViT feature inputs by completely overfitting base classes. The $c$, $\mathcal{B}$, and $\mathcal{C} \setminus c$ represents different groups of prototypes used in feature projection, as in Eq. 5. By projecting features to $p_c$, general detection ability emerges (nAP50: $4.5 \rightarrow 26.2$), then significantly improves after adding background classes prototypes $p_{\mathcal{B}}$ (nAP50: $26.2 \rightarrow 38.4$), and further improves with $p_{\mathcal{C} \setminus c}$ (nAP50: $38.4 \rightarrow 39.5$). Results in Tab. A2 and A3 are obtained on COCO 2017 with ViT-S/14.

**Region propagation network.** We study the impacts of region-propagation-based localization in Tab. A3. The conventional regression network exhibits poor localization accuracy on novel classes (nAP75: 14.6). Note that a set of learned 2d positional embeddings is added to the ViT features before feeding into the conventional regression network. When using our proposed region-propagation network, the localization accuracy has a significant boost (nAP75: $14.6 \rightarrow 24.1$). Since our region-propagation network expands proposals, the accuracy gain may come from the spatial features of larger areas. Therefore, we ablate this by expanding proposals within the conventional regression network. This results in even lower performance than the conventional regression network alone (nAP75: $14.6 \rightarrow 12.5$). This shows our region propagation network is the main contributing factor to the performance boost. Note that the results in Tab. A3 are obtained by only changing the localization architectures while keeping the feature subspace projection.

**Number of propagation layers.** We study the impacts of the number of PL blocks in Fig. A1, which shows that stacking more PL blocks consistently improves accuracy. The accuracy saturates around 5 blocks and there is still non-negligeble improvement from $3 \rightarrow 4$ blocks. Note that we use 3 blocks in our COCO and Pascal VOC experiments and 5 blocks in LVIS ones.

**Comparison to language-based detectors.** Tab. A4 compares our method DE-ViT against language-based detectors on the COCO and LVIS-v1 datasets. Language-based detection, also

Table A3: Ablation studies on the region propagation-based localization.

| Configuration | Novel | | Base | |
| --- | --- | --- | --- | --- |
| | nAP50 | nAP75 | bAP50 | bAP75 |
| Conventional regression network | 37.7 | 14.6 | 46.5 | 23.9 |
| Conventional regression network + expanded proposal | 35.6 | 12.5 | 41.3 | 19.8 |
| Region propagation network | **39.5** | **24.1** | 42.3 | 25.9 |

known as open-vocabulary detection [55], does not assume access to support images on novel classes but requires the knowledge of the novel class names. The class names are then used to discover images of novel classes from a large collection of image-text pairs. Therefore, language-based detectors have the following limitations. First, many objects lack clear names [56], *e.g.*, objects that are specific to certain contexts [57]. Second, the association between visual concepts and language is evolving and not static [58]. In contrast, few-shot object detection (FSOD) does not assume any knowledge of class names, and novel classes are described with support images only. FSOD's setting is therefore more general and arguably more challenging, and aims to emulate humans' capability to recognize objects by their consistent appearance.

Table A4: Comparison to language-based detectors on LVIS and COCO 2017. †: use customized pretrained backbone instead of public ones. "-": result that was not reported.

| Method | | Backbone | Use Extra Training Set | LVIS | | COCO |
| --- | --- | --- | --- | --- | --- | --- |
| | | | | mask APr | box APr | nAP50 |
| ViLD [18] | | EffNet-B7 | ✗ | 26.3 | 27 | 27.6 |
| RegionCLIP [32] | | RN50x4 | ✓ | - | 22 | 39.3 |
| OV-DETR [59] | | ViT-B/32 | ✗ | - | 17.4 | 29.4 |
| Detic [60] | | RN50 | ✓ | 17.8 | - | 27.8 |
| MM-OVOD [61] | | RN50 | ✓ | 25.8 | - | - |
| MM-OVOD (10-shots) [61] | | RN50 | ✓ | 27.3 | - | - |
| OWL-ViT [36] | | ViT-L/14† | ✗ | - | 25.6 | - |
| OWL-ViT [36] | | ViT-L/14† | ✓ | - | 31.2 | - |
| CORA+ [62] | | RN50x4 | ✓ | - | 28.1 | 43.1 |
| Ro-ViT [63] | | ViT-L/14† | ✗ | 31.4 | - | 33 |
| Co-Det [64] | | Swin-B | ✓ | 29.4 | - | 30.6 |
| F-VLM [63] | | RN50x64 | ✗ | 32.8 | - | 28.0 |
| DE-ViT (Ours) | 5 shots | ViT-S/14 | ✗ | 20.6 | 19.6 | 28.3 |
| | | ViT-B/14 | | 26.4 | 25.0 | 36.1 |
| | | ViT-L/14 | | 28.5 | 27.9 | 38.3 |
| | 10 shots | ViT-S/14 | | 21.8 | 21.9 | 36.1 |
| | | ViT-B/14 | | 29.9 | 28.1 | 42.4 |
| | | ViT-L/14 | | 32.4 | **31.9** | **46.2** |
| | 30 shots | ViT-S/14 | | 24.2 | 23.4 | 39.5 |
| | | ViT-B/14 | | 28.5 | 26.8 | **45.4** |
| | | ViT-L/14 | | **34.3** | **33.6** | **50** |

In COCO, our method DE-ViT outperforms the previous SoTA CORA+ by 6.9 AP50. Our method only trains on COCO while CORA+ uses ImageNet-21K [65] and COCO Captions [66] as additional training data. In LVIS, DE-ViT outperforms the previous SoTA on mask APr (+1.5 over F-VLM) and box APr (+2.4 over OWL-ViT). Note that we report the Co-Det [64] result with Swin-B backbone instead of EVA02-L backbone [67] because the latter includes the base and novel classes of COCO in its training set. Moreover, we observe a high variance in the performance of language-based detectors. F-VLM achieves a mask APr of 32.8 on LVIS but only has 28 nAP50 on COCO. While CORA+ has 43.1 nAP50 on COCO but only a box APr of 28.1 on LVIS. In contrast, our DE-

ViT outperforms existing solutions on both LVIS and COCO. MM-OVOD (10-shots) [10] denotes the MM-OVOD language detector with 10 images per novel classes provided as examples.

**Background classes set $\mathcal{B}$.** We study the impacts of using different background classes sets $\mathcal{B}$ in Tab. A6. The first and second rows represent using the stuff (non-object) classes, *e.g.*, floor, sky, of COCOstuff [34] and ADE20k [68] to construct $\mathcal{B}$. Note that COCOstuff has 53 stuff classes and ADE20k has 35 stuff classes, which explains the minor accuracy drop from ADE20k. The third row denotes using all classes of ADE20k as $\mathcal{B}$. This indicates that adding thing (object) classes to $\mathcal{B}$ may not be beneficial. Tab. A6 shows that changing $\mathcal{B}$ only results in small accuracy variations. This suggests that our proposed feature subspace projection does not rely on specific prototypes.

Table A5: Ablation studies on annotation types used to build prototypes.

| Support Images Annotation | nAP50 | | |
|---|---|---|---|
| | 5-shot | 10-shot | 30-shot |
| mask | 43.1 | 43.1 | 43.1 |
| bbox | 43 | 42.6 | 43.1 |

Table A6: Ablation studies on the background classes set $\mathcal{B}$.

| Background Classes $\mathcal{B}$ | nAP50 | bAP50 |
|---|---|---|
| COCOstuff [34] | 43.1 | 45.6 |
| ADE20k [68] | 42.5 | 45.8 |
| ADE20k (all) | 42.1 | 44.7 |

**More Shots.** We study the model performance with different numbers of shots in Fig. 7 and Fig. A2, on COCO 2014 and COCO 2017, correspondingly. For COCO 2014, the numbers of shots are set from 1 to 10, 15, 20, and 30. To align with existing work, we use the same support images by previous work [13] for shots 2,3,5,10,30. For other shots, we sample within the support images mentioned above. For COCO 2017, we follow the conventional base/novel class splits of the one-shot benchmark. To measure with more robustness, we randomly select support images within the validation set of COCO 2017 for each query image, and compute nAP50 for all four novel class splits. The reported nAP50 is the average among all splits and choices of support images. The numbers of shots are set from 1 to 10, 15, 20, 30, 40, 50, 75, 100.

Table A7: Comparison of training epoch and parameter size to other detectors trained on LVIS.

| | Total Params | Trained Params | Epochs | APr |
|---|---|---|---|---|
| OWL-ViT [36] | 433M | 433M | 1800 | 31.2 |
| F-VLM [69] | 445M | 25M | 118 | 32.8 |
| DE-ViT (Ours) | 350M | **23M** | **14.4** | **34.3** |

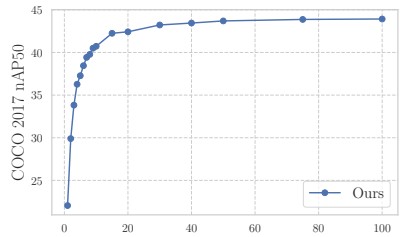

Figure A2: Detection accuracy under different numbers of shots in COCO 2017.

**Using Boxes or Masks to Build Prototypes.** We study the effects of annotation types such as bounding boxes or masks for prototype construction in Tab. A5. We observe that using bounding boxes to build prototypes yields almost indistinguishable performance compared to using instance masks at even 5-shot in COCO.

**Training Epochs and Parameter Sizes.** In Tab. A7, we compare the parameter sizes and training epochs of detectors trained on the large-scale dataset LVIS. Tab. A7 shows that our method DE-ViT only has 23M trainable parameters, and is trained orders of magnitude faster than F-VLM [69] and OWL-ViT [36].

**Hyperparameters.** We report the hyperparameters used in our experiments in Tab. A8.

Table A8: Hyperparameters used in our experiments.

| Hyperparameter | Value |
|---|---|
| Max image size | $800 \times 1333 \times 3$ |
| Batch size | 16 |
| Learning rate | 0.002 |
| Optimizer | SGD |
| Warmup steps | 5000 |
| Learning rate schedule | stepwise decay |
| Learning rate decay steps | 60000, 80000 |
| Learning rate decay gamma | 0.1 |
| Training steps | 90000 |
| Evaluation frequency | 5000 steps for COCO, 10000 steps for LVIS |
| Vision transformer size (layers, channel size) | small (12, 384), base (12, 784), large (24, 1024) |
| Channels of propagation network | 256 |
| Channels of feature subspace projection ($S$) | 256 |
| Classes in feature subspace projection ($T$) | 10 |
| Number of PL layers | 3 for COCO, 5 for LVIS |

**Preparation of YCB Objects.** Fig. 1 shows the detection results of DE-ViT on YCB objects, a standard set of objects widely used in robotic manipulation benchmark [6]. There are misclassifications and inaccurate boxes, *e.g.*, the white skillet is mistaken as a can, all round-shape fruits are recognized as orange, while the red one is an apple. However, we believe the overall result is encouraging. The specification of YCB objects at the time this paper is written includes 72 categories. We use a total of 33 by selecting and merging certain categories. The categories in use are `apple, ball, banana, bowl, brick, can, cheez-it, chips, clamp, cleanser bottle, coffee jar, comet pine, cups, drill, glass, lego, lemon, marker, mug, mustard, orange, peach, pear, peg-hole, pitcher, plate, screwdriver, skillet, spray bottle, sugar box, toy airplane box, utensil, wood blocks jar`. The source image in Fig. 1 is taken from the banner picture of ycbbenchmarks.com. For each category, we use Google Image Search to collect a few sample images. Fewer than four images on average are gathered per category. We annotate the corresponding objects by instance masks in each image using the software provided by SimpleClick [70]. Similar to SAM, SimpleClick generates instance masks automatically from user clicks, which significantly simplifies and accelerates the annotation procedure. Our annotator feedback indicates that annotating masks with SimpleClick is even easier and more accurate than drawing bounding boxes. An NVIDIA 3060 GPU is used for SimpleClick software. Class prototypes for YCB objects are built from the annotated example images. During DE-ViT inference, we replace prototypes of LVIS categories with those of YCB objects in order to detect these new categories. During postprocessing, We apply class-agnostic NMS and filter small bounding boxes. The data used for demonstration will be released upon acceptance.

# D  Limitations

In this work, we propose DE-ViT, a few-shot detector that uses example images to detect novel classes without any finetuning. We demonstrate that DE-ViT establishes new state-of-the-art in few-shot and one-shot benchmarks. One of the limitations is that our method DE-ViT uses a hybrid architecture of ViT and RCNN. The region proposal network (RPN) of RCNN is trained with only base classes and frozen during inference. This potentially limits the detection performance of novel objects. To overcome this limitation, one solution is to implement the RPN with our proposed framework as well. In doing so, the object proposals can be generated by conditioning on the prototypes of novel objects. Another solution is to adopt a full transformer architecture such as DETR [71] to eliminate the object proposal stage.

Our method DE-ViT adopts prototype-feature interactions, instead of dense spatial feature interactions that are computationally heavy. In doing so, our method DE-ViT archives greater accuracy and efficiency. However, class representative prototypes can potentially carry stronger bias from the pretrained ViTs such as DINOv2 than low-level spatial features. This would affect the model performance on long-tailed domains where the existing feature extractors underperform. A promising future direction is to design a more adaptive and bias-aware architecture to balance the efficient prototype-feature interactions and expensive spatial feature interactions.

