# OpenReview forum: "Detect Everything with Few Examples"
_robot-learning.org/CoRL/2024/Conference — CoRL 2024_

### Official Review · Reviewer_3v8u · 2024-07-17
**Novel FSOD algorithm demonstrated with pick-and-place task**

**Originality:** 4
**Technical Quality:** 4
**Clarity Of Presentation:** 3
**Potential Impact:** 3
**Recommendation:** 4
**Confidence:** 3

**Review:**

The novel FSOD framework introduced by this paper proposes interesting and notable solutions to localization or overfitting problems that arise in a naïve implementation (the paper notes these problems are mutually exclusive if ViT is frozen or not, line 29). The authors thoroughly compare accuracy on relevant benchmarks and note inference latency compared to other methods. The development of few-shot object detection algorithms is a highly relevant problem in robotics and an interactive demonstration in a realistic pick-and-place scenario is included in the paper.

The paper is generally well written and clear with two minor notes:
The subchapters of section 2 seem out of order; “feature subspace projection” could be explained before “propagation layer” since it is an input to the propagation layer.
In figure 3.3, Propagation layer has “input regions” of dimension 2×K×K, this should be changed to t×K×K to reflect the main body content on line 77 “The t-th PL block takes all previous regions…” to improve clarity of the figure.

**Quality Of The Limitations Section:**

3

**Questions For Rebuttal:**

Does Table A1’s secs/img include latency/cost of the lightweight prototype classifier to find the T classes (line 145)?

Were position embeddings (learned or sinusoidal) added post-hoc to the extracted ViT features for the “conventional regression” localization experiments (Table A3)?

Is ROI Align necessary in the “feature subspace projection”? Could the expanded proposal be aligned to pixels, rather than the original proposal box?

**Robotics Focus:**

3

**Summary Of Paper:**

This paper presents a novel few-shot object detection (FSOD) framework that achieves SOTA on various benchmarks and is demonstrated on YCB objects in a pick-and-place setting. The FSOD method uses an iterative proposal refinement process for localization and class prototypes to project pretrained ViT features into a subspace for processing and final classification. These prototypes can be of novel classes which enables few-shot object detection. This is achieved with novel “Propagation Layer” and “Feature Subspace Projection” modules, taking advantage of strong feature representations from a pretrained ViT.

**Summary Of Recommendation:**

This paper presents a novel few shot learning approach with strong performance on benchmarks. Furthermore, the authors create a framework for application in a pick-and-place setting demonstrating the viability of immediate application in robotics settings. However, there is a lack of a limitations section.

---

### Official Review · Reviewer_dSpH · 2024-07-20
**Initial Review of Few-Shot Object Detection without Fine-Tuning**

**Originality:** 3
**Technical Quality:** 4
**Clarity Of Presentation:** 4
**Potential Impact:** 3
**Recommendation:** 3
**Confidence:** 3

**Review:**

### Strengths
- The paper is clear and well-written
- The proposed approach is well-motivated and yields strong performance experimentally.
- The experimental analysis (if including all experiments in the supplementary materials) are rigorous and convincing demonstrate the performance of the method. It was interesting to see a comprehensive comparison to language vision models in the Appendix.
- The paper demonstrates the usefulness of the method in a real world robotics application, with a video demonstration in the supplementary materials.
- The code will be released as part of the submission


### Weaknesses
- The overall structure of the paper is non-standard and many key aspects are not in the main paper but rather the supplementary materials. For example, the comparisons to prior work is limited and important ablation experiments are only available in the supplementary.
- The related works section in the supplementary materials is short and not very comprehensive.
- The details of the channel re-order operation were not entirely clear. How exactly are the C-1 channels "interpolated" to a preset number of channels?
- The qualitative analysis of the robotics experiment is limited given only 10 trials for a few objects.

**Quality Of The Limitations Section:**

2

**Questions For Rebuttal:**

- What are the details of the channel re-order operation? (see weaknesses above)

**Robotics Focus:**

4

**Summary Of Paper:**

This paper presents a new approach to few-shot object detection without fine-tuning. The approach leverages a dot-product between  ViT features + prototypes to project features into a subspace relevant for the task. Importantly, the authors note that ViT features do not contain good information for regressing bounding boxes and therefore the authors propose a region propagation  layer that can be stacked to ultimately produce accurate 2d bounding boxes. The experimental analysis showcases the effectiveness of the approach on standard few-shot detection benchmarks compared to alternative approaches as well as vision-language detection models. Finally, the paper shows a downstream robotics application of the approach to sort new objects.

**Summary Of Recommendation:**

Overall, the paper presents an interesting new approach to few-shot object detection without fine-tuning. While I am not an expert in this area, the experimental analysis appears to be demonstrate the effectiveness of the method and I believe the paper, experiments and code would be useful to the community. I am open to revising my opining given other data, but for now I would argue for a weak accept.

---

### Official Review · Reviewer_dhu9 · 2024-07-22
**A good work in the field of few-shot learning, but a bit unrelated with robotics**

**Originality:** 3
**Technical Quality:** 4
**Clarity Of Presentation:** 4
**Potential Impact:** 3
**Recommendation:** 3
**Confidence:** 4

**Review:**

Strengths:
1. The paper proposes several innovative methods from the perspective of fine-tuning in few-shot, and the originality of the article is good. At the same time, each method has certain practical results in logic and experiments, which can prove that these methods are indeed effective for few-shot object detection and have good quality.
2.	The paper is clear in expression. It starts with the drawbacks of fine-tuning, and then explains the method based on this, and finally verifies it with experiments. The logic of the paper is relatively smooth.
3. The method mentioned in the article has achieved SOTA on multiple benchmarks, indicating that this working method has good effectiveness. At the same time, the method has also been experimented in fields such as open-vocabulary, showing that the method has certain effects in multiple fields.
Weaknesses:
1. The main focus of this paper is on the fine-tuning problem in few-shot object detection, which has been studied in many studies. At the same time, this work mainly focuses on model improvements, and its application in robot hardware and software is relatively limited. Although the paper conducts experiments on robot grasping, the contribution of this work is still more in computer vision rather than in robotics.
2. The paper is a little bit lacking in the description of the experimental settings, and the description of some experimental details is too brief, such as the number of epochs trained on COCO, batch size, etc.

**Quality Of The Limitations Section:**

1

**Questions For Rebuttal:**

1. There is an article named “One-shot object detection without fine-tuning”, which is also for the purpose of avoiding finetune, and can be compared with the results of this article if necessary. [1]
2. In your Table A4, the model using 30-shot is compared with open-vocabulary detectors. However, 30-shot is generally too much for vision-language models (e.g. [2] only used 10 visual examples for LVIS), so it’s better for you if you can provide experimental results with fewer shots and compare them to other methods in your Table A4.
3. In your Table A3, you conducted a series of ablation experiments. However, the experiments on individual parts need to be supplemented, such as solely Region Propagation Network.

[1] Li X, Zhang L, Chen Y P, et al. One-shot object detection without fine-tuning[J]. arXiv preprint arXiv:2005.03819, 2020.
[2] Kaul P, Xie W, Zisserman A. Multi-modal classifiers for open-vocabulary object detection[C]//International Conference on Machine Learning. PMLR, 2023: 15946-15969.

**Robotics Focus:**

2

**Summary Of Paper:**

The paper introduces DE-ViT, a novel few-shot object detection method that leverages pretrained vision transformers (ViTs) without the need for fine-tuning. The main idea is to address the limitations of traditional few-shot detection methods, which often require fine-tuning on both base and novel classes, leading to complex procedures and significant accuracy gaps between these classes. DE-ViT utilizes ViT features to detect novel objects based on a small set of support images, overcoming the need for fine-tuning by employing a dot-product with prototypes and a region propagation network to refine localization and derive class scores.  Experiment results show that DE-ViT achieves new state-of-the-art results on several benchmarks, including Pascal VOC, COCO, and LVIS, surpassing previous methods by significant margins in terms of both accuracy and inference time. It also demonstrates the effectiveness of DE-ViT in a real-world robotic task, showcasing its ability to detect and sort novel objects based on example images without any further training.

**Summary Of Recommendation:**

The overall logic of the paper is relatively clear, and a relatively complete method is designed based on the shortcomings of fine-tuning. In particular, the paper designed a large number of experiments, which achieved good results on datasets such as COCO and LVIS, and designed a series of ablation experiments to demonstrate the rationality of the module design. At the same time, this work also conducted experiments in the fields of actual robot operation and open-vocabulary object detection, and achieved good results. Therefore, this work has made good progress in the fields of robot perception and computer vision. But considering the relavant with robotics, I will not argue for my recommendation if other reviewers have a different opinion.

---

### Author Rebuttal · Authors · 2024-08-06

We are greatly thankful to the reviewers for the suggestions and acknowledgment of our work. We have addressed the concerns raised and uploaded the revised paper in the rebuttal file. Detailed responses can be found in the following individual comments.

---

### Decision · Program_Chairs · 2024-09-04

**Decision:**

Accept

**Comment:**

The paper proposes a new method for few-shot object detection that leverages pre-trained ViT models. A new region propagation network is designed to refine the location of region proposals.

Strengths

- The reviewers agree on the novelty of the proposed method for  few-shot object detection.
- The experimental results in the paper are convincing with comparisons to previous works.

Weaknesses
- The reviewers have raised several questions regarding clarifications of details in the paper and missing citations.

Post-Rebuttal
- The authors have successfully addressed the concerns from the reviewers.